# Knee Isokinetic Profiles and Reference Values of Professional Female Soccer Players

**DOI:** 10.3390/sports10120204

**Published:** 2022-12-12

**Authors:** Isabel Brígido-Fernández, Francisco García-Muro San José, Guillermo Charneco-Salguero, José Miguel Cárdenas-Rebollo, Yolanda Ortega-Latorre, Ofelia Carrión-Otero, Luis Fernández-Rosa

**Affiliations:** 1Facultad de Medicina, Universidad San Pablo-CEU, CEU Universities, Urbanización Montepríncipe, 28660 Boadilla del Monte, Spain; 2Departamento de Ciencias Básicas de Salud, Escuela Universitaria de Enfermería y Fisioterapia San Juan de Dios, Universidad Pontificia Comillas, 28350 Ciempozuelos, Spain

**Keywords:** isokinetic dynamometer, female soccer, reference values

## Abstract

Few studies have previously evaluated isokinetic parameters in female soccer players in comparison to those in males. The aim of this study was to describe normative quadriceps (Q) and hamstring (H) muscle strength values in professional female soccer players and to examine differences between dominant leg (DL) and nondominant leg (NDL). A standardized test protocol of concentric knee extension and flexion test protocol was conducted using the dynamometer isokinetic system (IsoMEd 2000). All the participants were healthy female professional soccer players from Spanish first and second division teams. Players were assessed for peak torque (PT) and maximum work (MW) values at 60°/s, 180°/s, and 240°/s. The mean difference was 7.17 (*p*-value = 0.0036), 4.4 (*p*-value = 0.0386), and 4.25 Nm (*p*-value = 0.0241) at speed 60°, 180°, and 240°/s, respectively. No statistically significant differences were detected for H–Q values between DL and NDL. This difference was 6.44 (*p*-value = 0.0449), and 5.87 J (*p*-value = 0.0266) at speed 60°, and 180°/s. The present study can be a tool that health professionals working with female professional soccer players in their care can use to assess and monitor a particular player.

## 1. Introduction

Soccer is the most popular sport in the world [1,2,3] and is performed by men and women, children, and adults with different levels of expertise. Women’s soccer has grown exponentially in recent years [4,5] and its participation rates are projected to increase to 60 million worldwide by 2026, doubling the current participation [4,6,7]. Soccer is a complex contact sport with high physical, technical, tactical, and physiological demands at the elite level [8,9]. Soccer requests a plethora of explosive athletic gestures, such as jumping, kicking, tackling, turning, sprinting, and changing pace, whose improvement is increased thanks to muscular strength training [10].

Muscle strength is not only affected by sex but is also sport-specific [11]. Thus, muscle strength has been defined as the maximum force or torque developed during maximal voluntary contraction under a given set of conditions and it can be also recorded in different contraction regimens, such as the most often applied isometric, but also the concentric and eccentric contraction regimens [12]. Players must be able of generating high torques during fast movements [2]. The more the force of muscular contraction is increased, the more the acceleration and speed will be increased in soccer’s turning, sprinting, and changing pace. Due to higher speed and aggressiveness of the current games, higher physical fitness levels and more intensive training are required, particularly at the professional level [8,9]. In this demanding environment, high levels of maximal strength in lower limbs can prevent injuries in soccer [1].

Isokinetic dynamometry (IKD) is considered the gold standard of strength assessments [13,14,15,16,17,18,19]. This measurement has been used by practitioners in professional soccer to quantify objectively the strength of the thigh musculature [20,21,22,23], their response to maximum intensity requirements, and the effective muscular capacity in processes of functional re-education [24]. Measures of isokinetic knee strength have been described throughout the literature to assess lower extremity strength basing on peak torque and maximum work [11,20,25,26,27,28]. Torque is defined by the rotational force about an axis that is equal to the product of a force times the distance from the axis where the force is applied [29]. The peak torque measurement is a method that is frequently employed for the objectification and evaluation of professional soccer players’ strength. Away from PT, there are described variables in isokinetic tests such as work [24]. The muscle strength of quadriceps (Q) and hamstrings (H) contributes significantly to lower limb biomechanics and athlete’s performance [25,30]. Traditionally, the focus of isokinetic research has been the measurement of absolute peak torque (PT) of Q and H [24,31]. The H–Q ratio based on peak torque has been traditionally used as a measure of strength imbalance [32]. Muscle strength deficits, such as between-leg asymmetries and imbalances between H and Q muscle strength (H–Q ratio), have shown to be important for determining readiness to return to sport [4,22,33]. Moreover, a lower H–Q ratio during concentric action has shown to increase the risk of lower-limb injuries [34]. Regarding maximum work (MW), few studies have examined the impact that work measurement can have on professional soccer players [24,35,36,37,38]. No references to muscle work were found in the literature about female soccer players. However, to assess isokinetic muscle endurance considering work as a predictor of that variable, MW values were taken into consideration. Work was defined as “the output of mechanical energy” [39] and represented by the area under the torque versus the angular displacement curve [40]. It has been consistently reported that young, adult, female athletes display differences in lower limb neuromuscular control and associated biomechanics, as well as Q and H muscle strength compared with their age and activity matched male peers [25]. It would be necessary to report torques with implications for performance and injury in females, since they show a higher risk of suffering thigh musculature and knee ligament injuries [20].

In professional female soccer, a continuous growth in research attention has been seen. However, the numbers of published articles are not comparable to those related to professional male soccer, [21,24,41,42,43,44] and science has struggled to keep pace with the demand for evidence-based studies as female players transition from the amateur to professional level [5]. A recent study [25] that included 196 female soccer players and evaluated peak torque at 60°/s demonstrated no significant differences in Q muscle strength between legs, but greater H muscle strength values were observed for the dominant leg compared with the nondominant leg. Despite this study described at first normative values, the authors asserted that reporting only isokinetic peak torque values can be considered as a limitation. For that, more publications including other variables that would help to form a more holistic profile of the female soccer players, by establishing reference values for muscle strength for athletes in specific sports, by age and gender is important to allow comparison of an individual’s values to his peers [25], but also reducing the risk of injury in these players [45,46], especially the anterior cruciate ligament (ACL), a lesion tremendously frequent in these athletes [38,47]. Although, there have been some studies on normative data from nonathletic populations [25,38] that included both men and women, the articles about female soccer players are scarce. Among them, several limitations, e.g., assessing a single angular velocity [25], considering a small sample number [11,48,49] or non-professional [50].

This study aimed to establish PT and MW in professional female soccer players and to examine differences in Q and H muscle strength between the dominant leg (DL) and the nondominant leg (NDL).

## 2. Materials and Methods

### 2.1. Participants

All the participants were healthy professional female soccer players from female soccer teams of the Spanish first (Liga Iberdrola) and second division (Reto Iberdrola), classified at the national and international level, who volunteered and took part in the study, which took place during the preseason of the soccer teams (August 2019). It is noteworthy to highlight that the recruitment period for participants was limited to that season due to the onset of the SARS-CoV-2 pandemic. To be included in this present study on normative muscle strength data for feminine soccer players, participants had to be female professional Federated soccer players as greater than or equal to 16 years old and with an experience higher than 4 years. Athletes were excluded if in the previously 6 months underwent any lower-extremity orthopedic surgery. Likewise, and despite being an innocuous test, it requires, however, a maximum effort for a short period of time. Therefore, and to avoid any type of fetal stress, pregnancy was another of the exclusion criteria in our study.

All participants were fully informed verbally and in writing about the study procedures prior to participation in the study and provided signed informed consent in accordance with the principles of the Declaration of Helsinki. The study was approved by the ethics committee of CEU San Pablo University (approval code: 345/19/19).

For players under the age of 18 years, signed informed consent was obtained not only from them but also from their parents to be eligible for participation.

After the soccer player’s arrival at the Laboratory of the Physical Therapies Research Unit at CEU San Pablo University, their affiliation was carried out and anthropometric data (height and body mass measurements) were obtained using a Mechanical Scale with Altimeter (Seca GmbH, Hamburg, Germany).

Each player also completed an individual information sheet with clinical data. As additional information: leg dominance, as well as the demarcation (goalkeeper, defender, midfielder, or striker) of each player were also collected. The dominant limb was determined by asking the players which limb they would choose for kicking a ball [48]. To ensure anonymity, each player’s identity was encrypted.

### 2.2. Sample Size Calculation

The sample size was calculated using Ene 3.0 software (Autonomic University of Barcelona, Barcelona, Spain). The calculation was based on the following formula:$$n=\frac{N*Z_{a}^{2}*S^{2}}{d^{2}*\left(N-1\right)+Z_{a}^{2}*S^{2}\;}$$
where n is sample size, N = 1062 Spanish professional female soccer players; $Z_{a}^{2}$ = 1.962 (confidence level set at 95%); d = 5; and $S^{2}$ is the variance of the quantitative variable (*dt*^2^) assumed to exist in the population. In relation to the *dt* of the variable of interest, this was given by preliminary results (unpublished data) on a sample of 15 professional female football players belonging to one of the clubs to be included in the study. The estimated desired sample size was calculated to be 68.

### 2.3. Procedures

Prior to isokinetic testing, participants underwent a 10-min standardized warm up on a cycle ergometer (Monark Exercise 818E, Vansbro, Sweden) at a self-selected moderate level as previous studies described [19,21,51,52], preventing potential fatigue [49] that would distort the isokinetic test data. Saddle height was adjusted to ensure correct cycling biomechanics and to allow a knee extension and a knee flexion. Isokinetic concentric knee extension and flexion muscle strength measurements were conducted using an IsoMed 2000 model isokinetic system (D&R FERSTL GmbH, Hemau, Germany).

Participants were assessed for peak torque, H–Q strength ratio, work and power values using isokinetic dynamometry. We used a standardized bilateral concentric/concentric continuous contraction program followed at low (60°/s), medium (180°/s), and high speeds (240°/s); knee extension and flexion in a useful range of motion ranged between 0 and 90° of knee flexion. These angular velocities have been previously used in several studies [11,25,38]. We asked participants to remove their shoes before sitting on the dynamometer. The players were tested in a sitting position with a hip 85°-angle (Figure 1). Subsequently, players were comfortably secured with padded straps around the pelvis, torso, and thigh. The first part of the test was performed around the right thigh, changing afterwards to the left one when that lower limb would be tested to minimize any compensatory movements. Femoral condyle of the tested limb was as an anatomical reference and aligned with the dynamometer axis of rotation following the manufacturer’s instructions. The arm of the dynamometer lever was fixed to the distal part of the tibia, settling a padded strap 2.5 cm over the medial apex malleolus [50], thus the entire leg could be aligned with the isokinetic lever that had to be moved. Before starting the isokinetic protocol, automatic static gravitational correction was applied according to the manufacturer’s procedures to adjust for its effect on torque. Players performed 5 consecutive contractions at low velocity (60°/s), 10 consecutive contractions at medium velocity (180°/s), and 25 consecutive contractions at high velocity (240°/s) with the right lower limb followed subsequently by the left.

The participants were required to perform these at maximum intensity and completing the full range of motion (ROM) of knee flexion and extension. Between sets, a minute of rest was allowed, where the thigh strap was allowed to loosen if required. This time of rest has been previously used by other studies [38,51,53]. Before each speed series, subjects were allowed to perform a couple of repetitions of submaximal knee extension/flexion to familiarize themselves with the dynamics required in the test at each moment. [48] During the isokinetic test, verbal encouragement and visual feedback with the screen were provided to all participants to help them focus on the quality and maximum intensity of their movement [24,54]. The break between changing the tested leg was about a minute.

### 2.4. Statistical Analysis

SPSS Statistics Version 27 program was employed for the statistical analyses. Data normality was evaluated by using a bilateral Lilliefors test. A two-tailed Student’s *t*-test was performed to compare PT, H–Q ratio, and MW values considering or not the equality of variances between the DL and NDL. The mean difference was estimated by subtracting the mean values obtained for DL minus the mean values obtained for NDL. A significance level of 0.05 was considered. The quantitative variables have been represented by their mean and standard deviation and the qualitative variables by their frequency.

## 3. Results

A total of 68 healthy professional female soccer players were included in the study. Participants’ characteristics are described in Table 1.

All data followed a normal distribution (*p*-value > 0.05 for Lilliefors test).

PT, H–Q ratio, and MW results are described in Table 2. Regarding PT values, a significant higher knee flexion PT was associated with the DL at all speeds evaluated (Table 2). The mean difference between the DL and NDL peak torque was 7.17, 4.4, and 4.25 Nm at speed 60°/s, 180°/s, and 240°/s, respectively. No statistically significant differences were detected for H–Q values between DL and NDL. Similarly, a significant higher knee flexion MW was associated with the DL at all speed evaluated (Table 2). The mean difference between the DL and NDL MW was 6.44 and 5.87 J at speed 60°/s and 180°/s, respectively, showing an inverse proportionality between the MW and the flexion speed. Knee extension MW values of DL and NDL showed non-significant difference at any speed. Table 3 shows normalized PT flexion and extension values. Regarding normalized PT values, a significant higher knee flexion PT was associated with the DL at all speed evaluated at speed 60°/s, 180°/s, and 240°/s (Table 3). Only the DL knee extension normalized PT value was significantly higher at speed 60°/s. 

## 4. Discussion

The aim of this study was to establish PT and MW in professional female soccer players and to examine differences in Q and H muscle strength between the dominant leg (DL) and the nondominant leg (NDL) considering low (60°/s), medium (180°/s), and high (240°/s) angular velocities. The main finding was a significant difference in the knee flexion peak torque between the dominant and nondominant legs and a significant difference in the knee flexion maximum work between the dominant and nondominant at speed 60°/s and 180°/s.

Isokinetic muscle strength testing at 60°/s has been usually used as a reference measurement for quadriceps muscle performances in healthy and ACL-injured patients [55]. This study describes the presence of a higher flexion peak torque of the dominant leg at 60°/s. This increase was 8.6% higher in the dominant legs. These differences are notably higher than those ones previously reported ranging between 2 and 5% [25,56]. This increase could be also detected at 180°/s and 240°/s. This differences of peak torque between the two legs are also corroborated even when the flexion peak torques were normalized to body weight (Table 3), which would be in line with authors previously reported in both female [25] and male soccer players [57]. This difference between dominant and nondominant legs would point to a higher hamstring strength in the dominant leg and would demonstrate, as expected, that the dominant leg responsible for kicking would be the strongest one [58]. This finding is in line with previous studies that have reported greater strength in the dominant leg in male soccer players [59,60] and disagrees with symmetry between players’ dominant and nondominant legs in young male soccer players [61,62] and female soccer players from 11 to 18 years old [63]. Interestingly, this finding may be related to the fact that female soccer players are more susceptible to injury to their ACL more often while standing on their nondominant leg [64].

Muscle strength deficits, e.g., imbalances between hamstrings-to-quadriceps (H–Q) ratios, have been found to be important for determining readiness to return to sport [30,65,66]. H–Q ratios ranging between 50 and 80% have been reported in many studies [25,43,67,68], although recent literature asserts that H–Q ratios values should be around 60% [69,70]. A lower than 60% H–Q ratio would point out a flexor muscles weakness related to the extensor muscles of the same limb, creating a muscular imbalance [38] and increasing the risk of suffering injuries [45,46,48]. Despite there being no differences in H–Q values between DL and NDL in this study, a directly proportional increase could be observed between the H–Q ratio and the angular velocity, which is consistent with the previously described by some authors who used angular velocities of 60°/s, 120°/s, 180°/s, and 300°/s [67] or only 60°/s, 180°/s, 240°/s [71], H–Q range between 55 and 64%, previously described by other authors in male and female soccer players [25,42,48]. Unlike, Holmcob et al. [71] reported a higher H–Q ratio in the NDL leg group of 12 female soccer players from a National Collegiate Athletic Association Division I university. The explanation given for this was that the NDLs producing higher H–Q ratios is directly related to the nature of soccer since this leg usually serves more as the stabilizing leg, by remaining stationary as the DL would be responsible for striking the ball [71].

An isokinetic dynamometer can measure maximum work. Maximum work has been proposed as the most sensitive isokinetic parameter for detecting differences between the different soccer players’ positions in the field [24]. Despite that, studies that have considered the impact of the maximum work measurement on professional soccer players are scarce [72]. Maximum work showed an inversely proportional relationship with angular velocity. Maximum work was significantly higher in the dominant leg at 60°/s and 180°/s than in the non-dominant leg.

This article is not exempt from at least four limitations. First, the studied group was composed of female soccer player from both first (n = 22) and second division (n = 46) and there may be significant differences in reported physical abilities across levels of play for female soccer players, as has been recently reviewed [73]. Second, the playing position has not been considered since the number per position was considered low. Third, knee dynamic valgus of the soccer players was not considered in this study. This would limit the compassion between female and male soccer players, since it has been reported that female athletes show four times greater activation of their hamstring muscles than males during dynamic stress gesture [74]. Fourth, the limb was not randomized. Although previous studies have randomized the limb [19,38,75], the bias caused by this limitation may not be large since the soccer players had enough rest between series and are used to the tests, hence there should be no learning or fatigue.

Furthermore, this study would offer some promising future perspectives, for instances, it would allow comparing the values described here with younger female soccer players or, even, with male soccer players by considering that the values described here can change because of age, BMI, playing position, among others.

## 5. Conclusions

In summary, the results of the present study can be a tool that health professionals working with female professional soccer players in their care can use to assess and monitor a particular player. Our study has a sample of professional female soccer players, a fact that differs from most articles published. Additionally, it studies the isokinetic profile in a more global way since it evaluates three different angular velocities. Finally, it has a study not only of peak torque but of maximum work, which provides a more holistic view of the player. This study aims to be one more step in the investigation of a growing population such as professional female soccer players, to achieve research comparable to that of their male counterparts.

## Figures and Tables

**Figure 1 sports-10-00204-f001:**
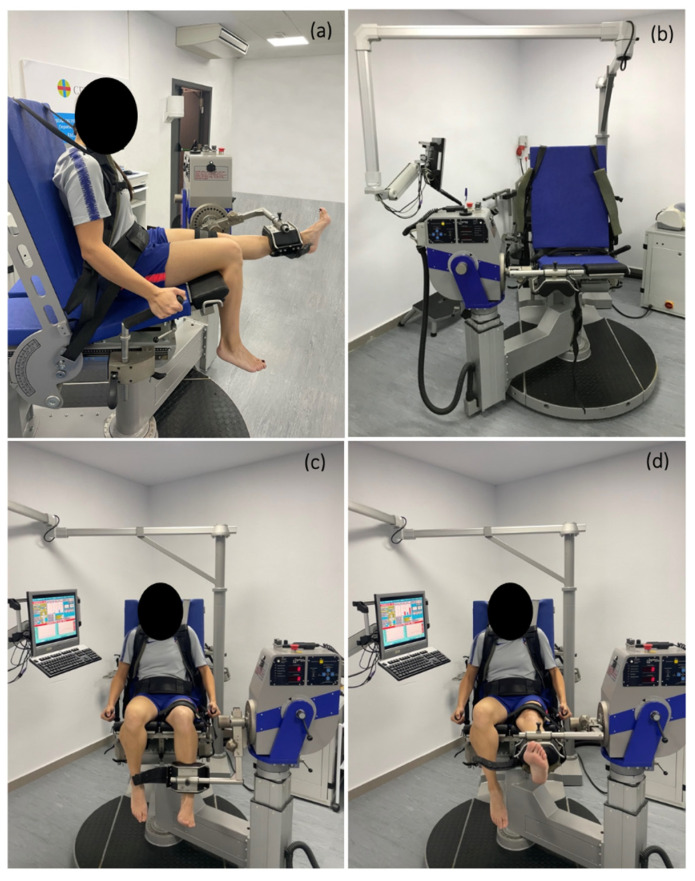
(**a**) Lateral view of the player with extension of the performing leg; (**b**) Frontal view of the Isokinetic dynamometer Isomed 2000; (**c**) Frontal view of the player with flexion of the performing leg; (**d**) Frontal view of the player with extension of the performing leg.

**Table 1 sports-10-00204-t001:** Participants’ characteristics.

**Participant Characteristics**	**Mean ± SD**
Age (years)	21.9 ± 4.19
Body height (m)	1.63 ± 0.05
Body mass (kg)	59.75 ± 6.19
Body mass index (kg/m^2^)	22.25 ± 1.6
**Leg dominance**	**Number of players (%)**
Right	60 (88.2)
Left	8 (11.8)
**Playing position**	**Number of players (%)**
Goalkeeper	9 (13.2)
Defender	17 (25)
Midfielder	22 (32.4)
Striker	20 (29.4)

**Table 2 sports-10-00204-t002:** Description of isokinetic PT, MW values during knee flexion and extension motion at different angular velocities (60°/s, 180°/s, and 240°/s). Means and SD obtained for the global sample (n = 68).

Knee Motion	Speed 60°/s					Speed 180°/s					Speed 240°/s				
	Overall ^$^	DL	NDL	*p*-Value	Cohen’s *d*	Overall	DL	NDL	*p*-Value	Cohen’s *d*	Overall	DL	NDL	*p*-Value	Cohen’s *d*
Peak torque (Nm)															
Flexion	83.29 ± 14.5 (79.84–86.73)	86.87 ± 14,96 (83.32–90.42)	79.7 ± 13.18 (76.57–82.83)	0.0036	0.5087	64.12 ± 12.44 (61.16–67.07)	66.31 ± 12.41 (63.36–69.26)	61.91 ± 12.15 (59.03–64.81)	0.0386	0.3579	57.60 ± 11.03 (54.98–60.23)	59.73 ± 10.71 (57.18–62.27)	55.48 ± 11.02 (52.86–58.10)	0.0241	0.3907
Extension	154.02 ± 23.39 (148.46–159.58)	158 ± 24.48 (152.18–163.81)	150.05 ± 21.71 (144.89–155.21)	0.0609	0.3975	111.65 ± 16.84 (107.65–115.66)	113.39 ± 18.19 (109.06–117.71)	109.92 ± 15.31 (106.28–113.56)	0.2309	0.2063	93.17 ± 14.18 (89.79–96.54)	94.88 ± 15.23 (91.26–98.50)	91.45 ± 12.93 (88.38–94.53)	0.1592	0.2435
H–Q ratio															
	0.54 ± 0.07 (0.56–0.52)	0.55 ± 0.07 (0.53–0.57)	0.53 ± 0.07 (0.52–0.55)	0.0981	0.2426	0.57 ± 0.09 (0.55–0.60)	0.58 ± 0.08 (0.57–0.61)	0.56 ± 0.09 (0.54–0.58)	0.1731	0.2517	0.62 ± 0.09 (0.59–0.64)	0.63 ± 0.08 (0.61–0.65)	0.61 ± 0.10 (0.58–0.63)	0.2001	0.2471
Maximum work (J)															
Flexion	107.63 ± 18.76 (103.17–112.09)	110.85 ± 18.40 (106.47–115.22)	104.41 ± 18.70 (99.97–108.86)	0.0449	0.3498	79.18 ± 15.49 (75.50–82.87)	82.12 ± 15.59 (78.41–85.82)	76.25 ± 14.94 (72.69–79.80)	0.0266	0.3844	62.58 ± 12.61 (59.58–65.57)	64.57 ± 13.08 (61.46–67.68)	60.58 ± 11.87 (57.76–63.41)	0.0647	0.3185
Extension	169.72 ± 28.24 (163.01–176.43)	173.37 ± 28.56 (166.58–180.15)	166.07 ± 27.65 (159.50–172.64)	0.1323	0.2607	131.43 ± 20.90 (126.46–136.40)	133.29 ± 22.16 (128.03–138.56)	129.57 ± 19.56 (124.93–134.22)	0.3012	0.1780	101.68 ± 16.33 (97.80–105.56)	103.48 ± 17.15 (99.41–107.56)	99.88 ± 15.39 (96.22–103.54)	0.1999	0.2211

Data are cited as mean ±SD (95% confidence interval). Abbreviations: DL, Dominant Leg; NDL, Nondominant Leg. *p*-values < 0.05 are represented in bold. $: This value describes each variable considering both legs.

**Table 3 sports-10-00204-t003:** Description of isokinetic normalized PT (Nm/kg) during knee flexion and extension motion at different angular velocities (60°/s, 180°/s, and 240°/s).

Knee Motion	Speed 60°/s			Speed 180°/s			Speed 240°/s		
	DL	NDL	*p*-Value	DL	NDL	*p*-Value	DL	NDL	*p*-Value
Flexion	1.47 (1.29–1.59)	1.3 (1.22–1.44)	**0.0005**	1.09 (0.97–1.2)	1.01 (0.91–1.11)	**0.0197**	0.99 (0.89–1.09)	0.91 (0.81–1.02)	**0.0074**
Extension	2.66 (2.46–2.84)	2.53 (2.37–2.72)	**0.0183**	1.88 (1.72–2.09)	1.82 (1.69–1.99)	0.1527	1.57 (1.43–1.76)	1.55 (1.39–1.64)	0.1901

Data are cited as median and interquartile range. Abbreviations: DL, Dominant Leg; NDL, Nondominant Leg. *p*-values < 0.05 are represented in bold.

## Data Availability

The data presented in this study are available on request from the corresponding author. The data are not publicly available due to ethical considerations.

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
