# Peer review of "Knee Isokinetic Profiles and Reference Values of Professional Female Soccer Players"

_sports, 2022, doi:10.3390/sports10120204_

Round 1

Reviewer 1 Report (Previous Reviewer 2)

Ref no 6: it doesn’t looks like a proper reference for that statement

Line no: 31 – How can the authors claim the current state with 14 year old reference

Line 52 – Is isokinetic considered as gold standard for muscle strength?

The introduction is not improved much.

The sample size was not calculated, and there was no upper age limit for participation.

The discussion is not proper. The authors should discuss each finding of their study 

Author Response

Reviever 1

We would like to thank the academic editor for having devoted time to read carefully, thoroughly evaluate and provide constructive criticism to our manuscript that will help to improve its overall quality one more time.

Ref no 6: it doesn’t look like a proper reference for that statement

It was a mistake. This reference has been corrected.

Line no: 31 – How can the authors claim the current state with 14 years-old reference

This statement has been modified and an updated reference has been placed instead.

Line 52 – Is isokinetic considered as gold standard for muscle strength?

We are really convinced that isokinetic dynamometry is the gold standard for assessing strength, therefore, we have further supported this statement with more references.

The introduction is not improved much.

We have modified the Introduction again, and we hope that the modifications satisfy the reviewer.

The sample size was not calculated, and there was no upper age limit for participation.

A new sub-section has been added to the Material and Method section, where we have described in detail how we estimated the sample size.

Regarding the upper age limit for participation, we have not set an upper age limit is that our inclusion criteria is that the player must be a professional, so if the player continues to be federated and has therefore been validated, being able to physically perform at the level required by professional soccer competition, any player over the age of 16 is valid for our research.

The discussion is not proper. The authors should discuss each finding of their study.

We have modified our Discussion section. We hope the modifications made satisfy the reviewer.

Reviewer 2 Report (New Reviewer)

Dear Authors, the manuscript is interesting. Perhaps, it lacks clinical implications, but it does provide data on a good sample of a professional population. In discussion I would expand with comparisons on your results and on those already present in the literature.

15 please remove the results of the sample selection from the objective

In the abstract methods, describe the population (major division, of a single team, single nation?)

In the outcome, as maximum work, do you mean the Maximum Voluntary Contraction?

35 athletic gestures instead of activity

In the introduction of the manuscript, I would describe the important value of strength in controlling the risk of injury, especially the ACL.

83 references missing

114 it would be interesting to compare minors and adults

168 have you taken the normality of the sample for granted?

Table 2 I suggest to insert the overall data

In addition to evaluating the comparison of minors and adults, it would be interesting to have a comparison with male professionals ..

“The altered neuromuscular timing and recruitment can lead to dynamic knee valgus stress in the lower limb  observed in women; it also reported that female athletes show four times greater activation of their hamstring muscles than males during dynamic stress gesture”

Ref: Marotta, N., Demeco, A., Moggio, L., Isabello, L., & Iona, T. (2020). Correlation between dynamic knee valgus and quadriceps activation time in female athletes. Journal of Physical Education and Sport, 20(5), 2508-2512 http://doi.org/10.7752/jpes.2020.05342

If you fail to imply these considerations, I would advise you to add in the study limitations.

Author Response

We would like to thank the academic editor for having devoted time to read carefully, thoroughly evaluate and provide constructive criticism to our manuscript that will help to improve its overall quality one more time.

Line 15 please remove the results of the sample selection from the objective

This result has been removed.

In the abstract methods, describe the population (major division, of a single team, single nation?)

This description has been added in the abstract methods.

In the outcome, as maximum work, do you mean the Maximum Voluntary Contraction?

No, we don’t. “Maximum Voluntary Isometric Contraction (MVIC) is an objective measure that has been adopted in the recent past to measure muscle strength, both in normal and in patients with neuromuscular disorders such as motor neuron disease, chronic inflammatory demyelinating polyneuropathy, post-polio, syndrome inclusion body myositis, spinal muscular atrophy, and facio-scapulo- humeral dystrophy. MVIC can be measured using a handheld dynamometer or, a strain gauge attached at one end to orthopaedic bars and at the other to the patient on a standard plinth. The force applied by the patient through the strain gauge creates a voltage, which is converted by computer into Newtons or kilograms. MVIC generates interval data which is more sensitive to change and more objective than manual muscle testing.” extracted from Meldrum et al. (Amyotroph Lateral Scler Other Motor Neuron Disord. 2003 Apr;4(1):36-44.)

The work indicates the capacity of the muscle to maintain the Force values throughout the entire range of motion. This maximum work offers us information about the maximum value of work developed in the series that we are measuring regardless of the repetition in which it is performed and is measured in Jules.

Line 35 athletic gestures instead of activity

This sentence has been modified.

In the introduction of the manuscript, I would describe the important value of strength in controlling the risk of injury, especially the ACL.

We would like to thank the reviewer this comment. We have included more information about the importance of this values in controlling the risk of injury in the Introduction section.

Line 83 references missing

Some references related to male soccer players evaluating isokinetic dynamometry have been added to this paragraph.

Line 114 it would be interesting to compare minors and adults

We agree with the reviewer, however we consider that the number of minor (n=12) was low to reach significant descriptive results compared to adults (n=56). Despite this, we have suggested this point as a potential future perspective related with this study in the Discussion section.

Line 168 have you taken the normality of the sample for granted?

No, we haven’t. We evaluated the normality of each variable at each speed. Now, we have expressed this explicitly in the Materials and Methods and Results section.

Table 2 I suggest inserting the overall data.

We have included in Table 2 that overall data and the 95% confidence intervals as another reviewer asked.

In addition to evaluating the comparison of minors and adults, it would be interesting to have a comparison with male professionals.

We agree with the reviewer, and we are delighted to share with the reviewer that we are working right now on that subject. Despite this, we have suggested this point as a potential future perspective related with this study in the Discussion section.

“The altered neuromuscular timing and recruitment can lead to dynamic knee valgus stress in the lower limb observed in women; it also reported that female athletes show four times greater activation of their hamstring muscles than males during dynamic stress gesture”. Ref: Marotta, N., Demeco, A., Moggio, L., Isabello, L., & Iona, T. (2020). Correlation between dynamic knee valgus and quadriceps activation time in female athletes. Journal of Physical Education and Sport, 20(5), 2508-2512 http://doi.org/10.7752/jpes.2020.05342. If you fail to imply these considerations, I would advise you to add in the study limitations.

This comment has been considered in the Discussion section as a limitation of our study.

Reviewer 3 Report (New Reviewer)

Dear Authors

You have written an interesting paper. However, some parts need to be addressed for greater clarity.

Abstract - initial sentence describing the main problem/goal in missing. Add

Add the model if the isokinetic dynamometer

Report p values

So what is the main conclusion of your paper? this is missing in the abstract. Amend

INTRODUCTION

The introduction does not report the most frequently used speeds for testing nor the range of motion or the PT/BW values from previous studies. No values that would report what good values for PT, work or H-Q ratio are. Please amend and add these values and don't be so descriptive.

METHODS

How was your sample size determined (G*Power od any other method )? Report

Why did you mic the sample between the 1st and 2nd leagues? This can impact your results. Elaborate and add in the limitations

Report the number of players per position.

Report their training experience

How did you determine the dominant leg? Report

You measured professional players. Why didn't you measure eccentric hamstring performance?

Self-selected pace? How did you make sure they were warmed up? What was the minimum setting of wats or RPM for the warm-up? Report

Why was the foot not chosen randomly? This can have an effect on your data. Elaborate and add to limitations.

Report exact ROM in degrees used.

Why a 1-minute break for all sets? 1 minute can be ok for 5 reps but too short for 10 reps. Elaborate and back up this with references.

What was the break between changing the leg? Report

Statistics - how was the normality of the data checked?

Report the 96% CI and effect size.

Why didn't you calculate any asymmetries as they are a key component in isokinetic measurements and football performance and predisposition for injuries? Add

Additionally, where is the PT to BW calculation? This is essential as PT without the BW does not mean much. ADD

Discussion

the discussion is poor and does not compare these values to the relevant literature.

H-Q ratio is a poor indicator and also the literature reports that it should be above 0.6-0.8.

In the discussion, you mention some differences/asymmetries however you don't report them in methods or in results.

The limitations are really modest and should be extended with all already highlighted above. Also, the values are generalised for all players and not by playing positions.

In general, this is a poorly written study. However, the data are promising. The whole structure and the lack of essential isokinetic-derived data are missing for this paper to be of any real practical value for practitioners and all involved in female football.

Kind regards

Author Response

We would like to thank the academic editor for having devoted time to read carefully, thoroughly evaluate and provide constructive criticism to our manuscript that will help to improve its overall quality one more time.

Abstract - initial sentence describing the main problem/goal in missing. Add

An initial sentence describing the main problem/goal has been added.

Add the model if the isokinetic dynamometer

The model of the isokinetic dynamometer has been added.

Report p-values

P-values have been added.

So what is the main conclusion of your paper? this is missing in the abstract. Amend

A conclusion has been added to Abstract.

INTRODUCTION

The introduction does not report the most frequently used speeds for testing nor the range of motion or the PT/BW values from previous studies. No values that would report what good values for PT, work or H-Q ratio are. Please amend and add these values and don't be so descriptive.

We have included Risberg et al. study in the Introduction. We follow this protocol using these speeds, because they are the ones used in our research unit, and we want to have homologous data to make, among other things, comparisons between the sexes with male peers in future studies. We have added some references in the Materials and Methods section that support us to have used these speeds (see Section 2.3 Procedures).

METHODS

How was your sample size determined (G*Power or any other method )? Report.

A new sub-section has been added to the Material and Method section, where we have described in detail how we estimated the sample size.

Why did you mic the sample between the 1st and 2nd leagues? This can impact your results. Elaborate and add in the limitations.

The first and second divisions, at least in the Spanish league, are professional leagues, in which the difference between players from one division and the other lies in their tactical quality, but not in their physical abilities, given that the workloads, hours of training and other physical requirements are equivalent. Due to the reviewer points that this may impact our results, we have included this point as a limitation of our study.

Report the number of players per position.

The demarcation (position) has been explicitly included in Table 1.

Report their training experience

Unfortunately, this piece of information was not asked during the interview. The only thing that we can reassure the reviewer is that their experience was higher than 4 years. Now, we have included this statement between the inclusion criteria.

How did you determine the dominant leg? Report

They were asked about their preferred kicking leg. This statement has been included in the Materials and Methods section and is supported by some reference where the authors did the same.

You measured professional players. Why didn't you measure eccentric hamstring performance?

Firstly, because when taking measurements in the preseason (recommended time for measurement in the literature), the players had just returned after their vacation period, so the level of muscle training is not in optimal training conditions and precisely for this reason we chose the conventional measurement method (which exclusively uses concentric contractions) in order to minimize any type of overexertion that could lead to some risk of injury at the start of the season, since eccentric contractions are more demanding. Second, since being part of a research group, previous studies on male players did not have eccentric contractions, and since our next objective is to compare data, we wanted to replicate the same protocol to be able to make comparisons between sexes. However, we understand that the information on eccentric contractions can be interesting, and we will take it into account for possible future study.

Self-selected pace? How did you make sure they were warmed up? What was the minimum setting of wats or RPM for the warm-up? Report

As professional athletes, they were supposed to have a sufficient level of knowledge to perform a basic warm-up on a stationary bike. However, they were verbally instructed to break a sweat during the bike warm-up before starting the isokinetic test. Subsequently, and prior to each speed test, a specific warm-up was done with 2-3 repetitions to warm up specifically.

Why was the foot not chosen randomly? This can have an effect on your data. Elaborate and add to limitations.

We agree with the reviewer’s comment, and we have included this point as a limitation of our studied in the Discussion section: “The foot was not randomized. Although previous studies have randomized the limb (Vargas VZ, et al. Knee isokinetic muscle strength and balance ratio in female soccer players of different age groups: a cross-sectional study. Phys Sportsmed. 2020.PMID: 31307251) the bias caused by this limitation may not be large since the soccer players had enough rest between series and are used to the tests, hence there should be no learning or fatigue.”.

Report exact ROM in degrees used.

The ROM was from 0º to 90º. This has been included in the Materials and Methos section (2.3 Procedure).

Why a 1-minute break for all sets? 1 minute can be ok for 5 reps but too short for 10 reps. Elaborate and back up this with references.

Because we followed the recommendation of previous studies. As requested, we have supported this statement by several reference.

What was the break between changing the leg? Report

It was about 1 minute, the time necessary to change the lever to one side, adjust the equipment with the anatomical references and re-cinch the player.

Statistics - how was the normality of the data checked?

We evaluated the normality of each variable at each speed. Now, we have expressed this explicitly in the Materials and Methods and Results section.

Report the 96% CI and effect size.

Now, we have included the 95% CI and the effect size (Cohen’s d) have been included in Table 2.

Why didn't you calculate any asymmetries as they are a key component in isokinetic measurements and football performance and predisposition for injuries? Add

We meant the differences between the dominant leg and nondominant leg peak torque and maximum work. Now, we have expressed this explicitly in the Results section.

Additionally, where is the PT to BW calculation? This is essential as PT without the BW does not mean much. ADD

It was considered to follow the Dr. Zeevi Devir’s recommendations in the Second Edition of “Isokinetics. Muscle testing, interpretation, and clinical applications”. On page 197. Strength normalization for bodyweight: “Therefore, the conclusion reached by the authors namely that bodyweight is the best normalizing factor for isokinetic angular and linear exertions is questionable. Consequently, it is recommended that in all instances, the absolute moment or force (linear lift) values should be quoted, leaving normalization to bodyweight an option”. However, we have added this point as a limitation of our study in the Discussion section.

On the other hand, most of the articles consulted (present and not in the bibliography of this work) and that address this issue offer their results in absolute values and few offer them standardized. As this is a controversial issue, we decided to follow the general recommendation.

Discussion

The discussion is poor and does not compare these values to the relevant literature.

The Discussion section has been modified.

H-Q ratio is a poor indicator and also the literature reports that it should be above 0.6-0.8.

We kindly disagree with the reviewer. We have rewritten the part related to H-Q ratio in the Discussion section considering the reviewer’s comment that H-Q ratio should be above 0.6-0.8.

In the discussion, you mention some differences/asymmetries however you don't report them in methods or in results.

Now, we have reported the origin of these differences in the Results section.

The limitations are really modest and should be extended with all already highlighted above. Also, the values are generalised for all players and not by playing positions.

The limitation paragraph has been extended and the limitation related to the playing position has been included.

In general, this is a poorly written study. However, the data are promising. The whole structure and the lack of essential isokinetic-derived data are missing for this paper to be of any real practical value for practitioners and all involved in female football.

We have broadly modified our manuscript and we hope that all the modifications are to the satisfaction of the reviewer.

Round 2

Reviewer 1 Report (Previous Reviewer 2)

The manuscript has been improved. I have only one comment 

Why the low sample size is a limitation, as you did a sample size calculation 

Author Response

We would like to thank the reviewer for having devoted time to read carefully, thoroughly evaluate and provide constructive criticism to our manuscript that will help to improve its overall quality one more time.

Why the low sample size is a limitation, as you did a sample size calculation

We agreed with the reviewer, for that, we have removed this limitation from the Discussion section.

Reviewer 2 Report (New Reviewer)

Dear authors, as I said earlier you provide a nice sample but a little insight for the literature. Because actually, when in line 13-14 "Few studies have previously evaluated isokinetic parameters in female soccer players in comparision to those in males". In fact, there are several manuscripts, with important datasets. (for example: https://www.tandfonline.com/doi/abs/10.1080/00913847.2019.1642808)

Anyway, the manuscript is clearer and more detailed, I would advise you to add correlation investigations, how the ratio changes v. age, v. BMI .. v. position to provide clinical implications to the literature.

I refer to the editor's considerations regarding the suitability of the manuscript, the latter methodologically appropriate.

Author Response

We would like to thank the reviewer for having devoted time to read carefully, thoroughly evaluate and provide constructive criticism to our manuscript that will help to improve its overall quality one more time.

Because actually, when in line 13-14 "Few studies have previously evaluated isokinetic parameters in female soccer players in comparision to those in males". In fact, there are several manuscripts, with important datasets. (for example: https://www.tandfonline.com/doi/abs/10.1080/00913847.2019.1642808).

This reference has been added in the Introduction section (line 67).

Anyway, the manuscript is clearer and more detailed, I would advise you to add correlation investigations, how the ratio changes v. age, v. BMI, v. position to provide clinical implications to the literature.

A comment considering the reviewer’s suggestion has been incorporated to the Discussion section.

Reviewer 3 Report (New Reviewer)

Dear Authors,

Thank you for addressing the majority of my questions. However, some parts still need to be addressed fully.

Why did you mic the sample between the 1st and 2nd leagues? This can impact your results. Elaborate and add in the limitations.

The first and second divisions, at least in the Spanish league, are professional leagues, in which the difference between players from one division and the other lies in their tactical quality, but not in their physical abilities, given that the workloads, hours of training and other physical requirements are equivalent. Due to the reviewer points that this may impact our results, we have included this point as a limitation of our study.

Really? Do they only differ in their tactical quality? And you can back up this with what research???

Self-selected pace? How did you make sure they were warmed up? What was the minimum setting of wats or RPM for the warm-up? Report

As professional athletes, they were supposed to have a sufficient level of knowledge to perform a basic warm-up on a stationary bike. However, they were verbally instructed to break a sweat during the bike warm-up before starting the isokinetic test. Subsequently, and prior to each speed test, a specific warm-up was done with 2-3 repetitions to warm up specifically.

Supposed? So how can someone replicate your study with a supposed warm-up rate? Please explain that to me in scientific terms. This needs to be added in the limitation section as the warm-up was clearly not supervised or planned adequately.

Report exact ROM in degrees used.

The ROM was from 0º to 90º. This has been included in the Materials and Methos section (2.3 Procedure).

Report ROM for isokinetic testing and not for the bicycle warm-up.

Why didn't you calculate any asymmetries as they are a key component in isokinetic measurements and football performance and predisposition for injuries? Add

We meant the differences between the dominant leg and nondominant leg peak torque and maximum work. Now, we have expressed this explicitly in the Results section.

Where did you highlight these asymmetries? It is not clear. Nor did you mention in the methods how did you calculate them.

It was considered to follow the Dr. Zeevi Devir’s recommendations in the Second Edition of “Isokinetics. Muscle testing, interpretation, and clinical applications”. On page 197. Strength normalization for bodyweight: “Therefore, the conclusion reached by the authors namely that bodyweight is the best normalizing factor for isokinetic angular and linear exertions is questionable. Consequently, it is recommended that in all instances, the absolute moment or force (linear lift) values should be quoted, leaving normalization to bodyweight an option”. However, we have added this point as a limitation of our study in the Discussion section.

On the other hand, most of the articles consulted (present and not in the bibliography of this work) and that address this issue offer their results in absolute values and few offer them standardized. As this is a controversial issue, we decided to follow the general recommendation.

With all respect to dr. Zeevi Dvir, this reference is from 2004 and a lot has changed in isokinetic since then. As you can see in the following paper the normalisation of PT to BW is essential in comparing the data between various research (https://www.frontiersin.org/articles/10.3389/fphys.2021.767941/full). Also in the literature, there is a call for standardisation of PT/BM from 2005 onwards (https://pubmed.ncbi.nlm.nih.gov/15903392/). Therefore, if you want to put out so-called reference values do a proper job or don't do it at all. Therefore, please report-add PT/BW values, as this would be an added value and could increase the citation value of your paper without authors doing extensive additional work.

References are done incorrectly - in-text citations count to 73, however, the reference list ends with 70. Please amend.

Overall the paper improved, however, there is still some work to do by the authors.

Author Response

We would like to thank the reviewer for having devoted time to read carefully, thoroughly evaluate and provide constructive criticism to our manuscript that will help to improve its overall quality one more time.

Why did you mic the sample between the 1st and 2nd leagues? This can impact your results. Elaborate and add in the limitations.

The first and second divisions, at least in the Spanish league, are professional leagues, in which the difference between players from one division and the other lies in their tactical quality, but not in their physical abilities, givÇ´ zen that the workloads, hours of training and other physical requirements are equivalent. Due to the reviewer points that this may impact our results, we have included this point as a limitation of our study.

Really? Do they only differ in their tactical quality? And you can back up this with what research?

We would like to thank the reviewer his/her comment. Stemming from this comment, we have found a recent article from Sports (Basel) where authors concluded that “there are significant differences in reported physical abilities across levels of play for female soccer athletes” (Sports (Basel). 2022 Oct; 10(10): 141.). Hence, we have included this among the limitations of our study.

Self-selected pace? How did you make sure they were warmed up? What was the minimum setting of wats or RPM for the warm-up? Report

 As professional athletes, they were supposed to have a sufficient level of knowledge to perform a basic warm-up on a stationary bike. However, they were verbally instructed to break a sweat during the bike warm-up before starting the isokinetic test. Subsequently, and prior to each speed test, a specific warm-up was done with 2-3 repetitions to warm up specifically.

Supposed? So how can someone replicate your study with a supposed warm-up rate? Please explain that to me in scientific terms. This needs to be added in the limitation section as the warm-up was clearly not supervised or planned adequately.

We kindly disagree with the reviewer. We have added further references supporting our warm-up methodology (lines 142-144). Our methodology is based on previous studies that described the rating of perceived exertion (RPE) (Borg. Med Sci Sports Exerc. 1982;14(5):377-81.; Blain et al., Int J Sports Physiol Perform. 2019 Jul 1;14(7):994-996. doi: 10.1123/ijspp.2018-0637.). This RPE can be used to monitor the exercise intensity during laboratory and specific tests, training sessions, and to estimate the internal training load of the athletes (Polito et al. Front Psychol. 2021 Jan 7;11:623480. doi: 10.3389/fpsyg.2020.623480. eCollection 2020.). From the physical perspective, the measure of the parameters which indicate the intensity of effort is necessary. RPE is a variable frequently used in the soccer as well, as it makes it possible for coaches and physical trainers to evaluate the intensity of the exercises exerted in a training session (Polito et al. Front Psychol. 2021 Jan 7;11:623480. doi: 10.3389/fpsyg.2020.623480. eCollection 2020.).

Report exact ROM in degrees used.

 The ROM was from 0º to 90º. This has been included in the Materials and Methos section (2.3 Procedure).

Report ROM for isokinetic testing and not for the bicycle warm-up.

We have rewritten this methodology in the Materials and methods section (lines 159-163): “Participants were assessed for peak torque, H-Q strength ratio, work and power values using isokinetic dynamometry. We used a standardized bilateral concentric/concentric continuous contraction program was followed at low (60º/s), medium (180º/s) and high speeds (240º/s), knee extension and flexion in a useful range of motion ranged between 0 and 90º of knee flexion. These angular velocities have been previously used in several studies [11,25,47].

Why didn't you calculate any asymmetries as they are a key component in isokinetic measurements and football performance and predisposition for injuries? Add

 We meant the differences between the dominant leg and nondominant leg peak torque and maximum work. Now, we have expressed this explicitly in the Results section.

Where did you highlight these asymmetries? It is not clear. Nor did you mention in the methods how did you calculate them.

We have added now an explicit sentence explaining the origin of this differences in the Materials and methods section (line 206-207).

Additionally, where is the PT to BW calculation? This is essential as PT without the BW does not mean much. ADD

 It was considered to follow the Dr. Zeevi Devir’s recommendations in the Second Edition of “Isokinetics. Muscle testing, interpretation, and clinical applications”. On page 197. Strength normalization for bodyweight: “Therefore, the conclusion reached by the authors namely that bodyweight is the best normalizing factor for isokinetic angular and linear exertions is questionable. Consequently, it is recommended that in all instances, the absolute moment or force (linear lift) values should be quoted, leaving normalization to bodyweight an option”. However, we have added this point as a limitation of our study in the Discussion section.

On the other hand, most of the articles consulted (present and not in the bibliography of this work) and that address this issue offer their results in absolute values and few offer them standardized. As this is a controversial issue, we decided to follow the general recommendation.

With all respect to dr. Zeevi Dvir, this reference is from 2004 and a lot has changed in isokinetic since then. As you can see in the following paper the normalisation of PT to BW is essential in comparing the data between various research (https://www.frontiersin.org/articles/10.3389/fphys.2021.767941/full). Also in the literature, there is a call for standardisation of PT/BM from 2005 onwards (https://pubmed.ncbi.nlm.nih.gov/15903392/). Therefore, if you want to put out so-called reference values do a proper job or don't do it at all. Therefore, please report-add PT/BW values, as this would be an added value and could increase the citation value of your paper without authors doing extensive additional work.

We agree with the reviewer, and taking the reviewer’s opinion into account, we have included a new Table 3 harboring this new variable and their pertinent statistical analyses. Furthermore, we modified the Discussion section for including these new results (lines 255-258).

References are done incorrectly - in-text citations count to 73, however, the reference list ends with 70. Please amend.

References have been checked and corrected.

This manuscript is a resubmission of an earlier submission. The following is a list of the peer review reports and author responses from that submission.

Round 1

Reviewer 1 Report

Introduction

In  general. Paragraph cannot comose of  a  single sentence .  I have seen a  lot of paragraphs  only  including 1 sentence (e.g., page 2, line 66, 77-79, 80-81, 82-83, 87-89, 92). Authors should put one topic per paragraph together.   

Page 1 line 30 to 37: You may  combine the first  and second  paragraph together. 

Page 1 line  43: It may be good to add examples of how physical fitness  and training intensity alter across years.  

Page 2 44 to 46: This paper discusses knee isokinetic profiles so the authors do not need  to include the paragraph from line 44 to 46 on page 2.

Page 2 line 47, page 3, line  115: Combine and restructure the sentences/paragraphs. You may start discussing  per paragraph as below

1. define muscle strength

2. How muscle strength is important in soccer.

3. How do you assess strength (e.g., Isokinetic testing)

4. measures (e.g, H-Q  ratio, Peak  Torque, maximum work)

5. What is the current issue in research: (based on  your claim, there are limited  reference values

The current issue I think is authors put explain muscle strength and the definition in a random order so it is very  difficult to read. The introduction should be composed of 6-7 paragraphs instead  of so many single sentence of paragraphs.

Page 3, line  117 to page  134: combine the sentences as one paragraph and include

1. current issue of research/literature

2. Why authors need to  research

3. What are the potential benefits of this study

4.  Purpose  statement

5.  Hypothesis

2. Methods

Page 3, line 142 to Page 4, line 153:  The sentences should be combined with the first paragraph.

Page 4, line 163: This should added after the Declaration of Helsinki.

Procedure: Do not create a paragraph if it is not necessary (e.g., page 4 line  171-172.

Page 4, line 174-175: modified to “IsoMed 2000 model isokinetic system (D&R FERSTL 174 GmbH, Hemau, Germany)”

Page 4, line 179-180: same as line 163.

Page 4, line 181-183: Reword this sentence.

Page 5, line 209-212: Based on results, the authors only provided descriptive statistics. Thus, why did you mention about p-value?

Results:

Overall:  Remove irrelevant sentences. Keep the section simple and concise. For example, in page 6, line 227-228, the authors can reword to “mean difference was 70.5 Nm between knee extensor and flexors in DL (p=xx)”.

Page 6 227-229: is the difference significant? Report p-value if so.

Page 6, line 235-238: rephrase them and remove irrelevant sentences.

Page 6, line 243-246: Same as above.

Page 250-252: Finally, you mentioned: “statistically significant”. Thus, if you said “difference” in the result sections, add  “significant or not significant” and p-value.

Page 6  line 253-261: In academia, you never used bullet points in results. Remove it. You should discuss in the discussion section.

Page 7 line 262-287:  The same principle would be applied as  3.2.

Page 7 line  289 to 298:  remove them.

Discussion:

Overall: authors  need to comprise of 3 to 5 paragraphs with similar topic. Current paragraphs are not sufficient to be published.

Page 8 line 311 – 320:  authors cannot leave 4 paragraphs of a single sentence in the discussion. Restructure them. Each sentence is not connected well at all.

Conclusion:

It’s well written compared to other sections.

Reviewer 2 Report

The introduction is poorly written

Most of the sentences in the introduction are repetitions

Line 177-93 – the same idea is repeated several times

What is the rationale for this study?

Why there is a lot of paragraphs with single sentences?

How the sample size was calculated? I have serious concerns regarding the sample size. The sample size looks very less.

What was the upper age limit for participation in the study?

Please add a figure for isokinetic testing

Line 215- don’t write ‘our’ players

Statistical analysis is not clear. Please use a separate subheading for statistical analysis

Was there any hypothesis tested in the study? why p-value

Line 300- 305 rewrite

The discussion is poorly written. Authors not following the standard format